# Curvature of the *Drosophila* corneal lens depends on localized chitin secretion

**Neha Ghosh, Eva Rojo-Iost¤, Jessica E. Treisman** [ID]*

Department of Cell Biology, NYU Grossman School of Medicine, New York, New York, United States of America

¤ Current address: Université Paris Cité, CNRS, Institut Jacques Monod, Paris, France
* Jessica.Treisman@nyulangone.org

## Abstract

The *Drosophila* corneal lens is an apical extracellular matrix structure with a biconvex shape that enables it to focus light onto the underlying photoreceptors. Here, we investigated how this shape is influenced by the source of one of its major components, the polysaccharide chitin. Knocking down the chitin synthase Krotzkopf verkehrt strongly reduced the thickness and curvature of the corneal lens. Conversely, enhancing chitin export by overexpressing Rebuf expanded and distorted the corneal lens. We found that the cone and primary pigment cells in the center of each ommatidium produce most of the chitin, and preventing chitin synthesis by these central cells reduced corneal lens curvature. Increasing chitin export from central cells increased the thickness of the central corneal lens, while increasing export from peripheral lattice cells made the edges thicker. The wild-type biconvex shape thus results from high levels of chitin production by central cells relative to peripheral cells, indicating that localized chitin secretion is critical for normal corneal lens curvature.

## Introduction

The shapes of most biological structures are primarily generated by the numbers, morphologies, and positions of their constituent cells. However, extracellular matrix (ECM) can also contribute to tissue and organ morphology. While basement membrane ECM has been shown to provide structural support to many epithelial cell types, the generation of diverse forms from apical ECM (aECM) is less well understood [1]. aECM structures include chitin-rich arthropod and collagen-rich nematode cuticles which act as permeability barriers and protect against dehydration, luminal scaffolds in tubular organs like the *Drosophila* trachea and the *C. elegans* excretory system, mechanosensory structures such as arthropod tactile hairs and mammalian whiskers, and the tectorial membrane of the vertebrate inner ear [2–5].

One aECM component that has been shown to play an important role in imparting shape to biological structures is the polysaccharide chitin, a polymer of N-acetylglucosamine [6]. For instance, newly synthesized chitin is required for the

**Data availability statement:** All relevant data are within the paper and its Supporting information files, including the source data file S1_Data.xls.

**Funding:** This work was funded by the National Institutes of Health (grant R01 EY035624 to J.E.T. and grant K99 EY037847 to N.G.). The funders played no role in the study design, data collection and analysis, decision to publish, or preparation of the manuscript. https://www.nih.gov/.

**Competing interests:** The authors have declared that no competing interests exist.

**Abbreviations:** APF, after puparium formation; aECM, apical ECM; BSA, bovine serum albumin; CBD, chitin-binding domain ECM, extracellular matrix; RT, room temperature.

luminal expansion of the fly embryonic tracheal system; in its absence, tracheal tubes form constrictions and cysts instead of expanding uniformly [7,8]. In *C. elegans*, a chitin layer provides structural rigidity to the eggshell, maintaining the oval shape of the embryo [9]. Chitin organization can be modified to form structures as diverse as lobster claws and structurally colored butterfly wing scales [10–12]. Chitin is a major component of insect corneal lenses [13–15], but its importance for the morphogenesis of these precisely curved structures is unknown.

The *Drosophila* corneal lens is an aECM structure primarily composed of chitin and chitin-binding proteins arranged in a biconvex shape that is essential to its function of focusing light onto the photoreceptors underneath. We have shown that corneal lens shape is compromised in mutants lacking the Zona Pellucida (ZP) domain-containing proteins Dusky-like (Dyl) or Dumpy (Dpy), and this shape change correlates with a delay in chitin deposition [16]. In addition, the external corneal lens surface is flattened in mutants lacking the transcription factor Blimp-1, the regulatory targets of which include genes that encode chitin processing enzymes such as Krotzkopf verkehrt (Kkv), Knickkopf (Knk), Rebuf (Reb), Expansion (Exp), Chitinase 5, Chitinase 7, and Chitinase 10 [17–21]. These observations prompted us to investigate the role of chitin in creating corneal lens shape.

In this study, we show that differential levels of chitin production between the central and peripheral cells of the ommatidium dictate the precise biconvex shape of the *Drosophila* corneal lens. Although all non-neuronal retinal cells express the chitin synthase gene *kkv*, the central cone and primary pigment cells are the major source of chitin for the thick middle region of the corneal lens, while the peripheral lattice cells contribute a small amount of chitin to the tapered corneal lens edges. We found that increasing chitin deposition by central cells by overexpressing Reb there increased the thickness of the middle region, making the corneal lenses more spherical. In contrast, increasing chitin deposition by lattice cells made the edges thicker, giving the corneal lenses a rectangular shape. These observations show that localized secretion and restricted diffusion of chitin establish the shape of the corneal lens.

## Results and discussion

### Chitin accumulates over central cells in the mid-pupal retina

To understand the role of chitin in corneal lens structure, we first characterized its production during corneal lens morphogenesis. Using a fluorescently labeled chitin-binding domain probe (CBD) [22] and super-resolution LIGHTNING microscopy, we examined the localization and organization of chitin in mid-pupal retinas. Chitin was not yet present at 48 h after puparium formation (APF) (Fig 1A). Chitin fibers first appeared over primary pigment cells at 50 h APF (Fig 1B), expanded to cone cells and became radially arranged at 51 h APF (Fig 1C), and were straighter and more tightly packed at 52 h APF (Fig 1D). By 54 h APF, chitin was evenly distributed in a dense meshwork that covered the apical surfaces of the cone and primary pigment cells (Fig 1E) and formed a thick layer with an outwardly curved shape (Fig 1F). However, no chitin accumulation was observed over the secondary and tertiary

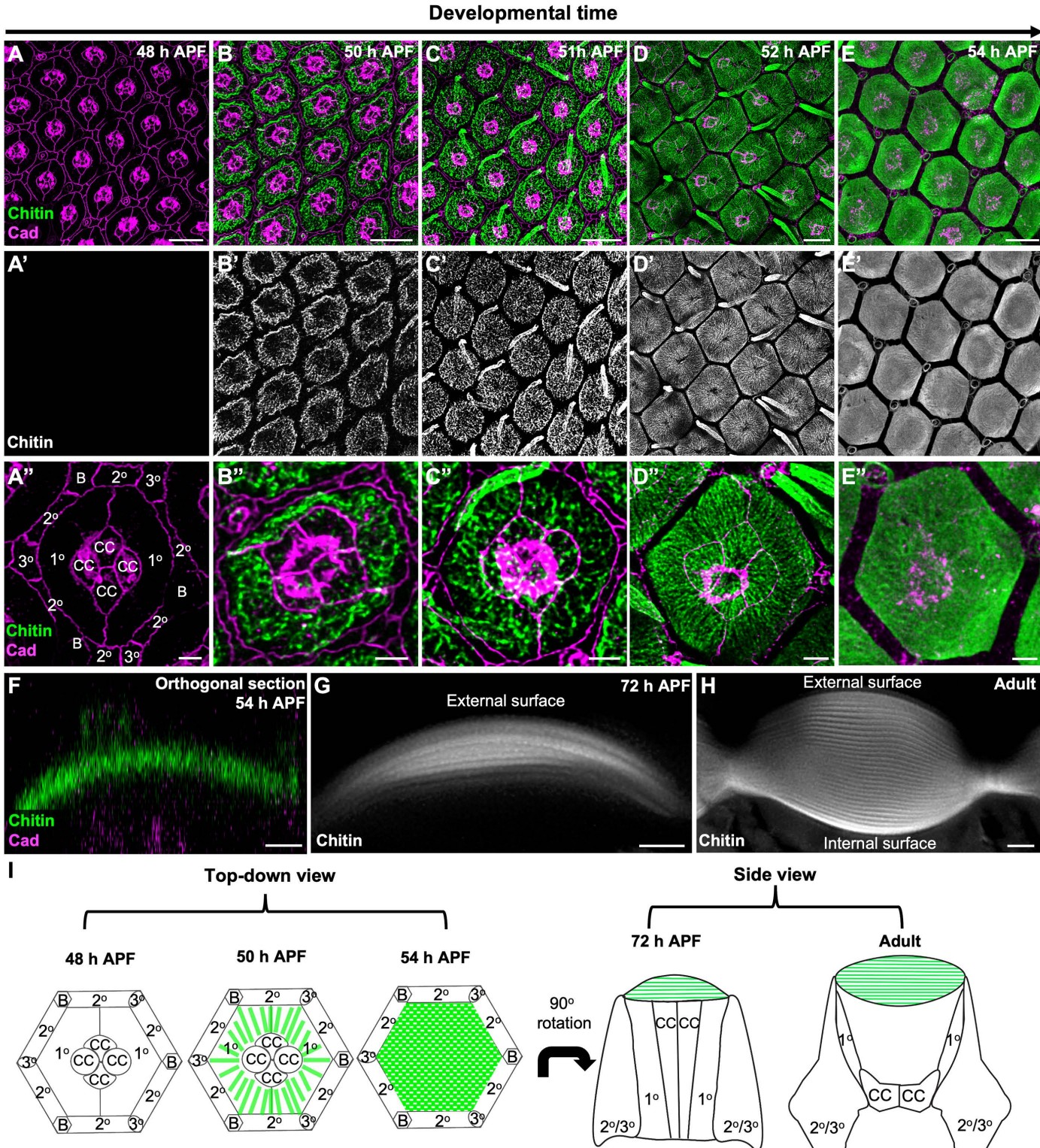

**Fig 1. Chitin organization during corneal lens morphogenesis.** (**A–E**) Apical surfaces of wild-type (*white^1118*) mid-pupal retinas stained for chitin with CBD (**A'–E'**, green in **A–E**) and E-Cadherin and N-Cadherin (Cad, magenta) at 48 h APF (**A**), 50 h APF (**B**), 51 h APF (**C**), 52 h APF (**D**), and 54 h APF

(**E**). (**A″–E″**) show enlargements of single ommatidia. (**F**) Orthogonal view of a single wild-type ommatidium at 54 h APF showing chitin (green) forming an externally curved layer apical to Cad (magenta). (**G, H**) Horizontal cryosections of single wild-type ommatidia showing 72 h APF (**G**) and adult (**H**) corneal lenses stained for chitin with Calcofluor White. Scale bars: 10 μm (**A–E**), 2 μm (**A″–E″**, **F–H**). (**I**) Diagram of chitin (green) organization during corneal lens development. CC, cone cells; 1°, primary pigment cells; 2°, secondary pigment cells; 3°, tertiary pigment cells; B, mechanosensory bristles.

pigment cells (lattice cells) at these stages. We examined chitin in horizontal cryosections of late pupal and adult corneal lenses by staining with Calcofluor White, a dye that penetrated these dense structures better than the CBD probe [16,21,23]. At 72 h APF, the corneal lens was externally curved [16] and had multiple chitin layers, especially in its central region (Fig 1G). The layers were more numerous and distinct in adult biconvex corneal lenses (Fig 1H). Initial chitin deposition thus occurs only over primary pigment cells and cone cells (collectively called central cells) during mid-pupal stages, and expansion of the proximal-distal axis of the corneal lens involves the progressive addition of chitin layers (Fig 1I).

### Disrupting chitin secretion alters corneal lens shape

We next investigated how chitin contributes to corneal lens architecture. The chitin synthase Kkv is an oligomeric transmembrane protein that polymerizes chitin chains and translocates them across the plasma membrane through a central channel [23,24] (Fig 2A). We blocked chitin synthesis by generating clones that expressed *kkv* RNAi with *IGMR-GAL4*, which is active in all retinal cells except the mechanosensory bristles [25]. Kkv antibody staining showed that RNAi expression effectively removed Kkv from the apical surfaces of non-neuronal retinal cells (Fig 2B). Clones expressing *kkv RNAi* showed no apically deposited chitin at 54 h APF, except in the bristles (Fig 2C), indicating that chitin is autonomously synthesized by underlying retinal cells and does not diffuse laterally. Adult corneal lenses produced by *kkv* knockdown ommatidia were completely devoid of chitin and were about half the thickness of wild-type corneal lenses, with reduced external and internal curvature (Figs 2D, 2E, 2N, 2O, 2P, and 3L). The level of Obstructor A (Obst-A), a protein that organizes the assembly of newly synthesized chitin and colocalizes with it in the larval epidermis and trachea [26,27], was also greatly reduced in these corneal lenses (S1A and S1B Fig). A second *Drosophila* chitin synthase, Chitin Synthase 2 (Chs2), is reported to be restricted to endodermal cells of the gut and inactive in ectodermal cells [28]. Consistent with this distribution and with the complete loss of chitin from *kkv* mutant ommatidia, we did not observe any effect of knocking down retinal *Chs2* on the chitin content or the shape of adult corneal lenses (S1C Fig).

Interestingly, a significant portion of the corneal lens was still present in the absence of chitin (Fig 2E). It is possible that some corneal lens components are retained by the transient external scaffold established by the ZP domain-containing proteins Dyl, Dpy, and Piopio (Pio) [16]. ZP-domain proteins can interact with aECM molecules in other contexts, making them good candidates to prevent the complete loss of the corneal lens. Pio links the apical plasma membrane to ECM in the embryonic trachea and attaches the embryonic cuticle to tendon cells, while Dyl plays a similar role in embryonic denticles and adult bristles [29–32]. We found that Dyl, Pio, and YFP-tagged endogenous Dpy were expressed normally in 51 h APF clones or retinas in which *kkv* was knocked down in all cells with *IGMR-GAL4* (S1D–S1G Fig), indicating that the scaffold is still formed in the absence of chitin. Although Dyl and Dpy are required for early chitin accumulation in the corneal lens at 54 h APF, chitin recovers at later stages of pupal development, when Dyl is no longer expressed. The adult *dyl* mutant phenotype is therefore distinct from loss of *kkv*; chitin is present in the inner portion of the corneal lens, which shows a deeper curvature than wild-type, and there are defects on the outer corneal lens surface [16]. If the scaffold is involved in retaining corneal lens material, its function must become dispensable late in development. Dpy-YFP is lost from the external surface and restricted to the pseudocone underneath the corneal lens by the adult stage [16], and the same change in localization was observed in *kkv* knockdown flies (S1H and S1I Fig).

The outermost layer of the corneal lens is composed of the Retinin protein and waxes [33], which may still be present in *kkv* knockdowns, as a similar envelope layer that surrounds olfactory sense organs is organized by Dyl and other ZP domain proteins rather than by chitin [34]. We were unable to test this possibility because we could not detect Retinin

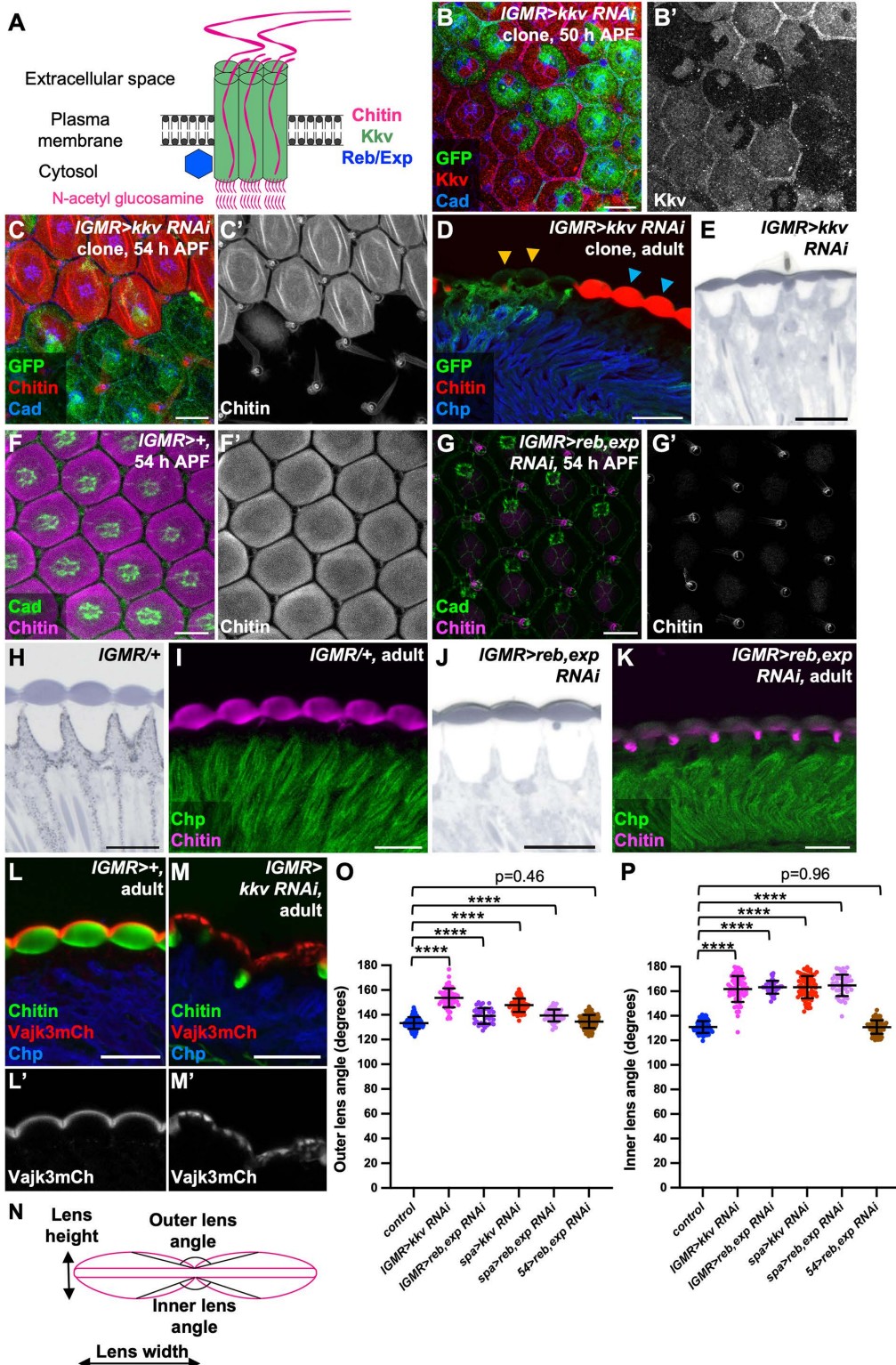

**Fig 2. Chitin production is necessary for biconvex corneal lens shape.** (**A**) Schematic showing the *Drosophila* chitin production machinery. (**B–D**) Retinas containing clones expressing *kkv* RNAi with *IGMR-GAL4* marked by GFP (green) and stained for Kkv (**B′**, red in **B**) or chitin (**C′**, red in **C**, **D**), and

Cad (blue, **B**, **C**) or Chaoptin, a marker for photoreceptor rhabdomeres (Chp, blue, **D**). (**B**) At 50 h APF Kkv is lost in *kkv* RNAi-expressing cells. At 54 h APF (**C**) and in adults (**D**) chitin is lost from the corneal lenses. Blue arrows mark wild-type ommatidia, and yellow arrows mark *kkv* knockdown clones (**D**). (**E**) Plastic section of an adult eye expressing *kkv* RNAi in all retinal cells with *IGMR-GAL4*, showing thin corneal lenses. (**F, G**) 54 h APF retinas stained for chitin (**F′, G′**, magenta in **F, G**) and Cad (green). (**F**) control *IGMR-GAL4/+*; (**G**) *IGMR-GAL4>reb RNAi; exp RNAi*. (**H–K**) Horizontal sections of control *IGMR-GAL4/+* retinas (**H, I**) or retinas expressing *reb RNAi* and *exp RNAi* in all retinal cells with *IGMR-GAL4* (**J, K**). (**H, J**) show plastic sections and (**I, K**) cryosections stained for chitin (magenta) and Chp (green). (**L, M**) Horizontal cryosections of pharate adult control *IGMR-GAL4/+* retinas (**L**) and retinas in which *kkv RNAi* is expressed in all retinal cells with *IGMR-GAL4* (**M**), stained for chitin (green) and Chp (blue) and showing the fluorescence of endogenously tagged Vajk3-mCherry (**L′, M′**, red in **L, M**). Vajk3-mCherry is still present, but less tightly localized, in the absence of chitin. Scale bars: 10 μm (**B, C, F, G**), 20 μm (**D, E, H–M**). (**N**) Schematic defining the outer and inner angles between adjacent corneal lenses, and the corneal lens height and width. (**O, P**) Graphs showing the outer (**O**) and inner (**P**) angles between adjacent corneal lenses in adult eye sections. Sample numbers (*n*) in this and subsequent figures are given as number of ommatidia/number of retinas. Wild-type control, *n* = 96 ommatidia/11 retinas; *IGMR>kkv RNAi*, *n* = 64/4; *IGMR>reb RNAi; exp RNAi*, *n* = 35/3; *spa>kkv RNAi*, *n* = 71/4; *spa>reb RNAi; exp RNAi*, *n* = 47/4; *54>reb RNAi; exp RNAi*, *n* = 113/9. The data underlying (**O, P**) can be found in S1 Data. For all graphs unless otherwise stated, error bars show mean ± SD, and *p* values were calculated using unpaired two-tailed *t* tests with Welch's correction and Bonferroni's correction for multiple comparisons. **** $p < 0.0001$.

on the outer surface of the adult corneal lens with an antibody that recognizes it on Western blots [33]. However, we did identify one corneal lens component that is still present in the absence of *kkv*. Four members of the Vajk family of cuticle proteins have been reported to bind to chitin in vitro [35]. We used CRISPR to tag the endogenous form of one of these proteins, Vajk3, with C-terminal mCherry. Interestingly, Vajk3-mCherry formed a defined layer external to chitin in wild-type corneal lenses (Fig 2L). This layer of Vajk3-mCherry was still present in *kkv* knockdown corneal lenses lacking chitin, although the protein distribution was less uniform and its boundary was less well-defined (Fig 2M). It is possible that Vajk3 is a component of an epicuticle-like layer [36] in the corneal lens that can form independently of chitin.

The chitin translocation activity of Kkv is supported by two interchangeable proteins encoded by the *reb* and *exp* genes [19,37] (Fig 2A). Reb and Exp contain Nα-MH2 domains and belong to the atypical SMAD group of proteins [38–40]. In the retina, Reb expression was observed primarily in lattice cells at 50 h APF (S2A Fig), while Exp was detected only in primary pigment cells at the same stage (S2C Fig), and both could be effectively depleted by RNAi (S2B and S2D Fig). By 54 h APF, Reb and Exp were both present in all central cells (S2E and S2F Fig). In the embryonic trachea, *reb* and *exp* act redundantly, and knocking down either gene alone does not affect chitin levels [19,37]. We found that *reb* knockdown in the retina with *IGMR-GAL4* did not alter chitin levels at 54 h APF or in adult corneal lenses (S2G and S2H Fig). Although *IGMR-GAL4*-driven *exp* RNAi caused a loss of apical chitin at 54 h APF, (S2I Fig), the corneal lens showed normal chitin accumulation by the adult stage and its shape was only slightly altered (S2J–S2L Fig). This suggests that Exp is the primary regulator of chitin secretion in the mid-pupal retina, while Reb compensates for loss of Exp at later developmental stages. Knocking down both *reb* and *exp* simultaneously in all retinal cells significantly reduced apical chitin deposition in both the 54 h APF retina and the adult corneal lens (Fig 2F–2K). Double knockdown of *reb* and *exp* resulted in corneal lenses with reduced height and decreased external and internal curvature, effects similar to those caused by depletion of *kkv* (Figs 3L and 2N–2P). Overall, these findings demonstrate that chitin constitutes and/or retains a significant portion of the volume of the corneal lens, consistent with a measurement of 20% chitin content by weight in the dragonfly corneal lens [13].

## Chitin production is primarily required in the central cells

As we observed initial chitin deposition only over the central cells, we hypothesized that these cells would be a major source of corneal lens chitin. Indeed, expressing *kkv* RNAi exclusively in clones of central cells using *sparkling-GAL4* (*spa-GAL4*) [41] eliminated nearly all chitin over the ommatidia at 50 h and 54 h APF (Figs 3A and S3A). A small amount of chitin remained in adult corneal lenses, which were abnormally shaped (Figs 3B, 3C, 3L, 2O, and 2P). In contrast, expressing *kkv* RNAi in clones of lattice cells using *54-GAL4* [42] did not affect chitin production at 50 h APF (S3B Fig) or its apical accumulation at 54 h APF (Fig 3G). There was a significant reduction in chitin intensity at the edges of the adult

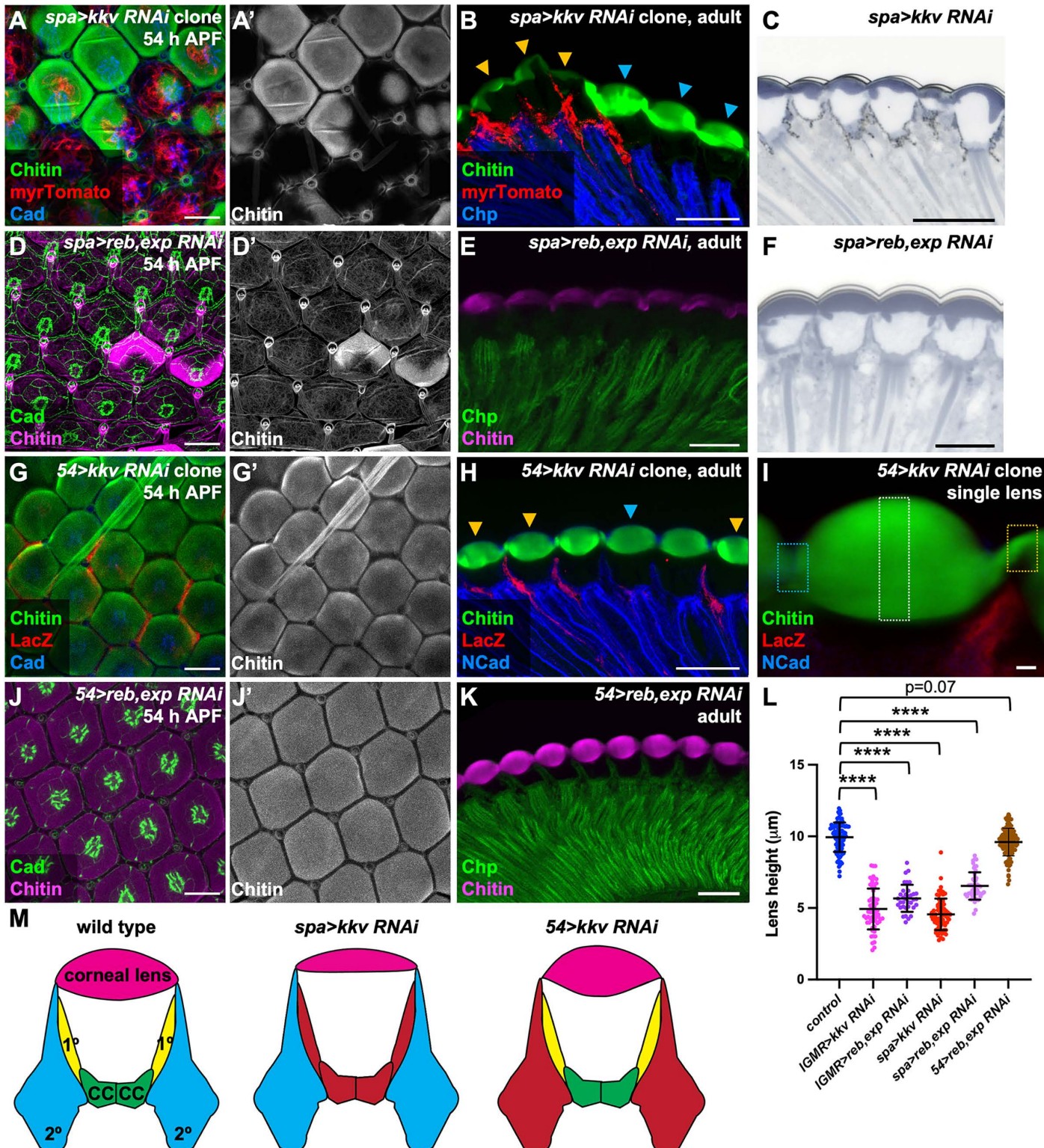

**Fig 3. Central cells are the major contributors of chitin.** (**A, D, G, J**) 54 h APF retinas; (**B, C, E, F, H, K**) adult eye sections. (**A, B**) *spa-GAL4* drives *kkv* RNAi expression in central cells in clones labeled with myrTomato (red), stained for chitin (**A′**, green in **A, B**) and Cad (blue, **A**) or Chp (blue, **B**). (**D,**

**E**) *spa-GAL4 > reb* RNAi, *exp* RNAi retinas stained for chitin (**D′**, magenta in **D, E**) and Cad (green, **D**) or Chp (green, **E**). (**C, F**) Plastic sections of *spa-GAL4 > kkv* RNAi (**C**) or *spa-GAL4 > reb RNAi, exp RNAi* (**F**) adult retinas. (**G–K**) Retinas in which *kkv RNAi* is expressed with the lattice cell driver *54-GAL4* in clones marked with anti-β-galactosidase (red, **G, H, I**) or *reb* RNAi and *exp* RNAi are expressed in all lattice cells with *54-GAL4* (**J, K**), stained for chitin (**G′, J′**, green in **G, H, I**, magenta in **J, K**), and Cad (blue in **G**, green in **J**), NCad (blue, **H, I**) or Chp (green, **K**). Blue arrows, wild-type ommatidia; yellow arrows, *kkv* knockdown clones (**B, H**). (**I**) shows an enlargement of a mosaic ommatidium indicating the regions in which chitin fluorescence intensity was quantified. Blue outline marks a wild-type corneal lens edge, yellow outline marks a *54-GAL4, UAS-kkv* RNAi edge, and white outline marks the central region used for normalization. Scale bars: 10 μm (**A, D, G, J**), 20 μm (**B, C, E, F, H, K**), 2 μm (**I**). Blue arrows, wild-type ommatidia; yellow arrows, *kkv* knockdown clones (**B, H**). (**L**) Graph showing the maximum height of corneal lenses in adult eye sections for wild-type control (*n* = 96/11), *IGMR>kkv RNAi* (*n* = 64/4), *IGMR>reb RNAi; exp RNAi* (*n* = 35/3), *spa>kkv RNAi* (*n* = 71/4), *spa>reb RNAi; exp RNAi* (*n* = 47/4), and *54>rebRNAi; exp RNAi* (*n* = 113/9). The data underlying (**L**) can be found in S1 Data. (**M**) Model depicting the effects of removing chitin from specific cell types. The cells that express *kkv* RNAi are colored red.

corneal lens (Figs 3H, 3I, and S3D), indicating that lattice cells do secrete some chitin later in development. Although lattice cells express both Kkv and Reb at 50 h APF, and these factors are sufficient for chitin secretion by ectodermal cells [19], it is possible that at this stage they lack another factor that is essential for chitin production, or that they express high levels of chitinases.

We also tested the effects of blocking chitin translocation by knocking down *reb* and *exp* in either central or lattice cells. Knocking down both *reb* and *exp* simultaneously in central cells significantly reduced apical chitin deposition at 54 h APF and in adult corneal lenses (Fig 3D–3F), resulting in thinner corneal lenses with reduced internal and external curvature (Figs 2O, 2P, 3F, and 3L). However, *reb* and *exp* double knockdown in lattice cells only reduced chitin intensity at the edges of the adult corneal lens, and had no significant impact on earlier chitin accumulation or on corneal lens shape (Figs 2O, 2P, 3J–3L, S3C, and S3D). Central cells thus appear to be the primary chitin-producing cells in the retina, and chitin production by these cell types is essential for normal corneal lens size and shape (Fig 3M).

## Localized production of excess chitin alters corneal lens shape

We next investigated the consequences of excessive chitin production. As previously reported in the trachea, where Kkv controls chitin polymerization but not its translocation across the membrane [19,37], we found that misexpression of *UAS-kkv* was not sufficient to increase chitin deposition or alter adult corneal lens structure (S4A–S4F Fig). However, misexpression of *reb* can promote excess chitin deposition by *kkv*-expressing ectodermal cells [19,37]. We found that overexpressing *reb* in clones of all retinal cells with *IGMR-GAL4* led to premature production of disorganized chitin fibers at 48 h APF (Fig 4A) and a large excess of apical chitin accumulation by 54 h APF (Fig 4B). Adult corneal lenses had distorted shapes, with chitin appearing to leak into the underlying pseudocone (Fig 4C). Chitin entered the pseudocone early in its development, at 84 h APF, and lamellar chitin organization in corneal lenses was disrupted (S4G Fig). The abnormal organization of chitin fibers in the presence of excess Reb implies that their organized assembly into curved lamellar structures (Fig 1) is a controlled process that requires gradual chitin addition, consistent with the requirement for organizing proteins such as Obst-A and Knk [43]. The inner curvature of the corneal lens is especially sensitive to excessive and disorganized chitin deposition, perhaps because the envelope layer helps to maintain the outer curvature.

To determine how corneal lens shape was affected by the cellular source of excess chitin, we overexpressed *UAS-reb* in clones of either central or lattice cells. *reb* overexpression in central cells caused premature chitin deposition at 48 h APF and excessive chitin accumulation in the center of corneal lenses at 54 h APF (Fig 4D and 4E). The chitin fibers formed tufts that extended apically, instead of being organized radially. In contrast, overexpressing *reb* in lattice cells resulted in very modest increases in chitin at 48 h and 54 h APF (Fig 4G and 4H), perhaps because Reb is already present in wild-type lattice cells at 50 h APF (S2A Fig). Adult corneal lens shape was altered by both manipulations. Clones in which central cells misexpressed *reb* produced more spherically shaped corneal lenses with increased height, reduced width, and increased outer and inner curvature, a change that was already apparent at 84 h APF (Figs 4F, 4J–4M, and S4H). The layered structure of chitin was disrupted in the inner part of the expanded corneal lens (Fig 4F). In contrast,

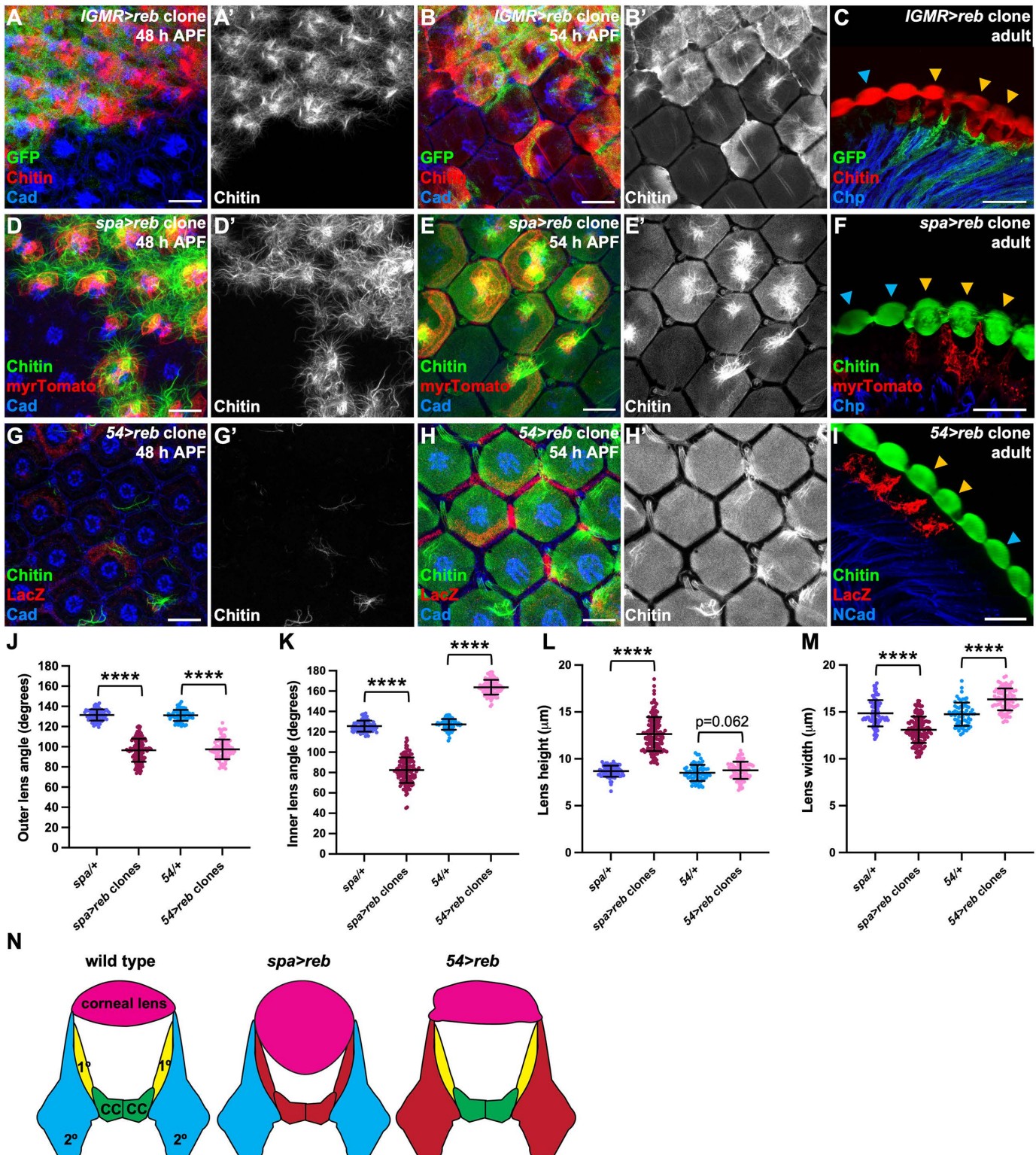

**Fig 4. Overproduction of chitin by specific cells alters corneal lens shape.** (A–C) Misexpression of *reb* with *IGMR-GAL4* in clones marked by GFP (green). (D–F) Misexpression of *reb* with *spa-GAL4* in clones marked by myrTomato (red). (G–I) Misexpression of *reb* with *54-GAL4* in clones marked

by *lacZ* (red). Retinas are stained for chitin (**A′, B′, D′, E′, G′, H′**, red in **A–C**, green in **D–I**) and Cad (blue in **A, B, D, E, G, H**), Chp (blue in **C, F**), or Ncad (blue in **I**). (**A, D, G**) 48 h APF; (**B, E, H**) 54 h APF; (**C, F, I**) adult. Scale bars: 10 µm (**A, B, D, E, G, H**), 20 µm (**C, F, I**). Blue arrows mark wild-type ommatidia and yellow arrows *reb*-expressing clones (**C, F, I**). (**J–M**) Graphs showing the outer (**J**) and inner (**K**) angles between adjacent corneal lenses, corneal lens height (**L**) and corneal lens width (**M**) in adult eye horizontal cryosections for wild-type control regions and clones in which *UAS-reb* is driven by *spa-GAL4* (*n* = 79/16 wild-type, 122/16 *spa-GAL4>UAS-reb*) or *54-GAL4* (*n* = 63/12 wild-type, 92/12 *54-GAL4>UAS-reb*). The data underlying (**J, K, L, M**) can be found in S1 Data. (**N**) Model showing the effect of *reb* misexpression in specific retinal cell types on corneal lens structure. The cells that express *reb* are colored red.

clones in which lattice cells overexpressed *reb* produced rectangular corneal lenses with expanded rather than tapered edges and flattened internal surfaces (Fig 4I–4M). Thus, our results suggest that corneal lens shape is determined by differential production of chitin by central and lattice cells. There appears to be little lateral mobility of chitin, so high levels of chitin production by central cells expand the center of the corneal lens, while lower levels of chitin produced by lattice cells later in development create the tapered corneal lens periphery. Excessive chitin from either source can alter adult corneal lens curvature (Fig 4N).

## Controlling chitin secretion and mobility can impart shape to aECM structures

Our study illustrates how differential production of the polysaccharide chitin by specific cell types can sculpt an aECM structure. The cone and primary pigment cells are specialized to secrete large quantities of chitin necessary for the expansion of the central corneal lens, while peripheral lattice cells secrete lower levels of chitin later in development, producing the narrower edges of the corneal lens. The limited ability of lattice cells to contribute chitin to the corneal lens may be due to their lack of *exp* and only transient expression of *reb* at mid-pupal stages. Alternatively, the paucity of chitin deposition by lattice cells might be due to higher expression of chitinases or lower expression of proteins such as Knk that can protect chitin from degradation [44]. We previously showed that expansion of the apical surfaces of central cells is necessary for them to produce or retain chitin early in pupal development [16], suggesting that the narrow apical surfaces of lattice cells might also restrict their ability to deposit chitin. Consistent with this possibility, the cells that produce chitinous scales in Manduca undergo endoreplication to expand their size [45]. Apical expansion of central cells into a domed shape [46] may initiate the curvature of the external corneal lens surface, which is still partially retained in the absence of chitin. Even when produced in vast excess, chitin is unable to spread to neighboring cells, indicating that it must be effectively trapped by plasma membrane and/or ECM proteins. We have identified the chitin-binding protein Vajk3 [35] as one potential chitin-trapping protein, as it is localized external to the chitin layer independently of the presence of chitin. Our observations extend previous work showing that localized secretion contributes to the morphogenesis of other apical ECM structures such as the taenidial folds of the *Drosophila* trachea and the tectorial membrane of the mammalian inner ear [47–49].

Chitin fibers can undergo self-assembly into higher-order crystalline structures stabilized by non-covalent interactions [6,50]. In the insect cuticle, microfibrils consisting of 17–20 antiparallel α-chitin polymers are aligned to form horizontal laminae [51]. Our observations confirm that the *Drosophila* corneal lens is also constructed of chitin microfibrils that are stacked into distinct layers to expand its height. These layers are less well resolved at the edges of the corneal lens, where they may be thinner and/or more tightly packed. In the dragonfly corneal lens, chitin appears to be organized differently on the outer and inner surfaces, with longer fibers on the outer surface [13]. The inner part of the *Drosophila* corneal lens, which is synthesized relatively late in development, may also prove to be distinct from the outer part. The changes in corneal lens shape that we observe on modulating chitin deposition by cone and primary pigment cells imply that the thickness of the central corneal lens directly depends on the amount of chitin produced by these cells. It is possible that the local concentration of chitin regulates layer thickness and/or the initiation of new layers. The packing of chitin layers could also be influenced by Knk or other chitin-binding proteins. Mass spectrometric analysis of the fly corneal lens identified only four major proteins [52], but similar studies in mosquito [53] and transcriptomic analyses of the pupal and adult

retina [21,54] suggest that the corneal lens contains many additional cuticle proteins, such as Vajk3, which may organize chitin and limit its lateral diffusion.

Due to its nontoxic and biodegradable properties chitin has found numerous applications in biomedicine. For instance, chitin-containing hydrogels are used in corneal transplants [55,56] and as dressings to promote wound healing [57,58], deacetylated chitin (chitosan) nanoparticles constitute safe drug delivery systems [59], and nanocomposite fibers containing chitin are used as scaffolds for bone engineering [60]. Understanding how chitin scaffolds for biological structures such as the corneal lens are established may provide insights relevant to these biotechnological applications of chitin [61].

## Materials and methods

Table of materials used (Table 1)

### Fly stocks and genetics

*Drosophila melanogaster* strains were maintained on standard yeast-cornmeal-agar media and raised at 25 °C. For analysis of the pupal retina, white prepupae (0 h APF) were collected with a soft wetted brush and cultured at 25 °C till the appropriate developmental stage. Both sexes were used interchangeably for all the experiments, as no sex-specific differences were observed.

*Drosophila melanogaster* stocks used to generate *kkv* knockdown clones were: (1) *kkv RNAi TRiP.HMC05880; FRT82B/SM6.TM6B*; (2) *UAS-CD8-GFP, ey3.5-FLP; lGMR-GAL4; FRT82, tub-GAL80/SM6-TM6B* (3) *ey-FLP, spa-GAL4; UAS-myr-Tomato; FRT82, tub-GAL80* (4) *UAS-CD8-GFP, ey3.5-FLP; 54-GAL4, UAS-lacZ; FRT82, tub-GAL80/SM6-TM6B*. Stocks used to knock down *kkv*, Chs2, *reb* and/or *exp* or to overexpress *kkv* were: (1) *kkv RNAi TRiP.HMC05880*; (2) *reb RNAi TRiP.HMJ21908*; (3) *exp RNAi TRiP.HMS01445*; (4) *reb RNAi; exp RNAi*; (5) *lGMR-GAL4*; (6) *spa-GAL4*; (7) *54-GAL4, UAS-lacZ/ SM6-TM6B*; (8) *Dpy-YFP; lGMR-GAL4*; (9) *Vajk3-mCherry; lGMR-GAL4*; (10) *Chs2 RNAi TRiP.HMJ22450;* (11) *UAS-Kkv-GFP*. Stocks used for *reb* overexpression in clones were: (1) *UAS-reb, FRT82B*; (2) *UAS-CD8-GFP, ey3.5-FLP; lGMR-GAL4; FRT82, tub-GAL80/SM6-TM6B* (3) *ey-FLP, spa-GAL4; UAS-myr-Tomato; FRT82, tub-GAL80* (4) *UAS-CD8-GFP, ey3.5-FLP; 54-GAL4, UAS-lacZ; FRT82, tub-GAL80/SM6-TM6B*.

Stocks were obtained from the following sources: *UAS-reb, UAS-kkv-GFP* [37], *kkv RNAi* (BL65006), *reb RNAi* (BL57861), *exp RNAi* (BL35032), *Chs2 RNAi* (BL35032): Bloomington *Drosophila* Stock Center (BDSC). *Dpy-YFP* (Kyoto115238): Kyoto Stock Center.

To tag Vajk3 with mCherry, *nanos-Cas9* ZH-2A flies (BL54591, BDSC) were co-injected with the pCFD5 plasmid containing gRNA1 (GTAGTACGATCATAACAACG) and a *Vajk3* sgRNA predicted to induce cleavage close to the stop codon (CCATTCCGTCCGCTTACTTG) and with a donor pUC-GW-Kan-based plasmid in which the mCherry coding sequence immediately followed the last amino acid of Vajk3 and was flanked by 100 bp each of upstream and downstream *Vajk3* genomic sequence and by gRNA1 target sites [63]. Plasmids were synthesized by Azenta (https://www.genewiz.com) and the injection was done by GenetiVision (https://www.genetivision.com). Injected adults were crossed to *yellow, white; CyO/Sco* flies, and the F1 generation were screened for red fluorescence in the eyes and body. Six lines were recovered from two independent F0 flies and all showed the same pattern of fluorescence. In-frame fusion of Vajk3 to mCherry was confirmed by PCR and sequencing.

### Immunohistochemistry

For cryosectioning, adult or pupal heads with the proboscis removed were glued onto glass rods using nail polish and fixed for 4 h in 4% formaldehyde in 0.2 M sodium phosphate buffer (pH 7.2) (PB) at 4 °C. The heads were then incubated through an increasing sucrose gradient in PB (5%, 10%, 25%, and 30% sucrose) for 20 min each, transferred to plastic molds containing TFM compound and frozen on dry ice. Cryosections of 12 μm were cut at −21°C, transferred onto positively charged slides, and a hydrophobic boundary was drawn around the sections with ImmEdge Hydrophobic

 

PLOS Biology

**Table 1. Materials used.**

| REAGENT or RESOURCE | SOURCE | IDENTIFIER |
|---|---|---|
| **Antibodies** | | |
| Chicken polyclonal anti-GFP | Thermo Fisher Scientific | Cat# A-10262RRID: AB_2534023 |
| Rabbit polyclonal anti-DsRed | Takara Bio | Cat# 632496RRID: AB_10013483 |
| Chicken polyclonal anti-β-galactosidase | Abcam | Cat# ab9361RRID: AB_307210 |
| Rat monoclonal anti-ECad | Developmental StudiesHybridoma Bank (DSHB) | #DCAD2RRID: AB_528120 |
| Rat monoclonal anti-NCad | DSHB | #MNCD2RRID: AB_528119 |
| Mouse monoclonal anti-Chp | DSHB | #24B10RRID: AB_528161 |
| Mouse monoclonal anti-Arm | DSHB | #N2 7A1RRID: AB_528089 |
| Rabbit polyclonal anti-Kkv | [37] | N/A |
| Rabbit polyclonal anti-Reb | [19,37] | N/A |
| Rat monoclonal anti-Exp | [19,37] | N/A |
| Rabbit polyclonal anti-Dyl | [31] | N/A |
| Rabbit polyclonal anti-Pio | [62] | N/A |
| Rabbit polyclonal anti-Obst-A | [43] | N/A |
| Goat anti-chicken 488 | Invitrogen | A-11039RRID: AB_2534096 |
| Donkey anti-chicken Cy3 | Jackson ImmunoResearch | 703-165-155RRID: AB_2340363 |
| Goat anti-rabbit Cy3 | Jackson ImmunoResearch | 111-165-003RRID: AB_2338000 |
| Donkey anti-mouse 488 | Invitrogen | A-21202AB_2534069 |
| Donkey anti-mouse Cy3 | Jackson ImmunoResearch | 715-165-151RRID: AB_2315777 |
| Donkey anti-rat Cy3 | Jackson ImmunoResearch | 712-165-150RRID: AB_2340666 |
| Donkey anti-rat Cy5 | Jackson ImmunoResearch | 712-175-153RRID: AB_2340672 |
| **Chemicals, peptides, and recombinant proteins** | | |
| Chitin Binding Domain-Alexa Fluor 488 | [16,21] | N/A |
| Chitin Binding Domain-Alexa Fluor 546 | [16,21] | N/A |
| Calcofluor White (Fluorescent Brightener 28 disodium salt solution) | Sigma Aldrich | Cat# 910090 |
| 10% Formaldehyde | Polysciences | Cat# 04018−1 |
| 20% Paraformaldehyde | Fisher Scientific | Cat# 50-980-493 |
| Bovine Serum Albumin (BSA) | Millipore Sigma | A4737 |
| Tissue Freezing Medium (TFM) | GeneralData | TFM-C |
| ImmEdge Hydrophobic Barrier Pen | Vector Laboratories | Cat# H-4000 |
| **Experimental models: Organisms/strains** | | |
| *D. melanogaster: w^1118* | Bloomington Drosophila Stock Center (BDSC) | BDSC:3605; FlyBase:FBst0003605 |

*(Continued)*

**Table 1.** (Continued)

| REAGENT or RESOURCE | SOURCE | IDENTIFIER |
|---|---|---|
| D. melanogaster: IGMR-GAL4 | [25] | N/A |
| D. melanogaster: spa-GAL4 | [41] | BDSC:26656; FlyBase:FBal0127274 |
| D. melanogaster: 54-GAL4 | [42] | Flybase: FBti0012688 |
| D. melanogaster: kkv RNAi TRiP.HMC05880 | BDSC | BDSC:65006; FlyBase: FBti0184093 |
| D. melanogaster: reb RNAi TRiP.HMJ21908 | BDSC | BDSC:57861; FlyBase: FBti0164346 |
| D. melanogaster: exp RNAi TRiP.HMS01445 | BDSC | BDSC:35032; FlyBase: FBti0144968 |
| D. melanogaster: UAS-reb | [37] | Flybase: FBal0302994 |
| D. melanogaster: UAS-CD8-GFP, ey3.5-FLP; IGMR-GAL4; FRT82, tub-GAL80/SM6-TM6B | [21] | N/A |
| D. melanogaster: ey-FLP, spa-GAL4; UAS-myr-Tomato; FRT82, tub-GAL80 | [21] | N/A |
| D. melanogaster: UAS-CD8-GFP, ey3.5-FLP; 54-GAL4, UAS-lacZ; FRT82, tub-GAL80/SM6-TM6B | [21] | N/A |
| D. melanogaster: kkvRNAi; FRT82B | This study | N/A |
| D. melanogaster: rebRNAi; exp RNAi | This study | N/A |
| D. melanogaster: UAS-reb; FRT82B | This study | N/A |
| D. melanogaster: Chs2RNAi TRiP.HMJ22450 | BDSC | BDSC: 58321;FlyBase: FBti0164715 |
| D. melanogaster: DpyYFP | Kyoto Stock Center | Kyoto115238; FlyBase:FBti0143891 |
| D. melanogaster: UAS-kkv-GFP | [37] | N/A |
| D. melanogaster: Vajk3mCherry | This study | N/A |
| D. melanogaster: Vajk3mCherry; IGMR-GAL4 | This study | N/A |
| D. melanogaster: DpyYFP; IGMR-GAL4 | This study | N/A |
| **Software and algorithms** | | |
| Image J | National Institutes of Health | http://imagej.nih.gov.ij/ |
| Adobe Photoshop | Adobe | |
| GraphPad Prism 10.0 | GraphPad Software | https://www.graphpad.com/ |
| MS-Office (MS-Word, MS-Powerpoint, MS-Excel) | MS-Office | |
| **Other** | | |
| Confocal Microscope | Leica SP8 II | |
| Super resolution LIGHTNING Microscope | Leica Stellaris 8 | |

Barrier Pen (Vector Laboratories). The slides were then postfixed in 0.5% formaldehyde in PB at room temperature (RT) for 30 min, washed in PBS with 0.3% Triton X-100 (PBT) three times for 10 min each, blocked for 1 h at RT in 1% bovine serum albumin (BSA) in PBT and incubated in primary antibodies overnight at 4 °C in 1% BSA in PBT. After three 20-min washes in PBT, slides were incubated in secondary antibodies in 1% BSA in PBT for 2 h at RT and mounted in Fluoromount-G (Southern Biotech). A 1:10 dilution of Calcofluor White solution (25% in water; Sigma Aldrich, 910090) was included with the secondary antibodies where indicated.

Pupal retinas attached to the brain were dissected from staged pupae and collected in ice-cold PBS in a glass plate. These samples were fixed on ice in 4% formaldehyde in PBS for 30 min. The samples were washed three times for 10 min each in PBT and incubated overnight at 4 °C in primary antibodies in 10% donkey serum in PBT. After three 20-min washes in PBT, the samples were incubated for 2 h in secondary antibodies in PBT/10% serum at RT and washed three times again for 20 min in PBT. Finally, the retinas were separated from the brain and mounted in 80% glycerol in PBS.

The primary antibodies used were: mouse anti-Chp (1:50; Developmental Studies Hybridoma Bank (DSHB), 24B10), chicken anti-GFP (1:400; Thermo Fisher, A-10262), chicken anti-LacZ (1:1000; Abcam, ab9361), rat anti-Ecad (1:10, DSHB, DCAD2), rat anti-Ncad (1:50, DSHB, DN-Ex), rabbit anti-Kkv (1:300) [37], rabbit anti-Reb (1:100) [19,37], rat anti-Exp (1:100) [19,37], rabbit anti-Dyl (1:300) [31], rabbit anti-Pio (1:300) [62], rabbit anti-ObstA (1:100) [43]. All antibodies were validated either using mutant or knockdown conditions as shown or by verifying that the staining pattern matched previously published descriptions. The secondary antibodies used were from either Jackson ImmunoResearch (Cy3 or Cy5 conjugates used at 1:200) or Invitrogen (Alexa488 conjugates used at 1:1000). Fluorescently labeled SNAP-CBD-probes (1:200) [16,21] were included with the secondary antibodies for staining pupal retinas. Images were acquired on a Leica SP8 confocal microscope for normal confocal microscopy or on a Leica Stellaris for super-resolution LIGHTNING microscopy with a 63X oil immersion lens and processed using ImageJ and Adobe Photoshop.

## Plastic sections

Adult heads of *Drosophila* were dissected and fixed in a freshly made fixative containing 2% paraformaldehyde, 2.5% glutaraldehyde, and 0.05% Triton X-100 in 0.1 M sodium cacodylate buffer (pH 7.2) at room temperature for 4 h and then overnight at 4 °C. The fixed heads were rinsed with 0.1 M sodium cacodylate buffer and post-fixed with 1% $OsO_4$ in 0.1 M cacodylate buffer, followed by dehydration in a graded ethanol series (30%, 50%, 70%, 85%, 95%, 100%), infiltrated with propylene oxide/Spurr mixtures and finally embedded in Spurr resin (Electron Microscopy Sciences, PA, USA). Semithin horizontal 1 µm sections were cut and mounted on a glass slide, then baked on a hot plate overnight at 37 °C. The sections were stained with 0.1% Toluidine Blue, dried on a hot plate, and imaged with a Zeiss Axioplan microscope and processed in Adobe Photoshop.

## Quantification and statistical analysis

The outer and inner angles between adjacent corneal lenses were measured according to the schematic in Fig 2N, using the angle tool in ImageJ. To measure lens height, freehand straight lines were drawn in the center of the corneal lens from the upper to lower surface using the line tool in ImageJ and measured. Total chitin fluorescence was measured in a rectangle 3 µm wide centered on the junction between two corneal lenses and normalized to the fluorescence in a rectangle of the same width centered on the midpoint of the corneal lens (Fig 3I). Values were plotted in GraphPad Prism v10. Significance was calculated using Welch's two-tailed unpaired t-tests with Bonferroni correction for multiple comparisons when necessary. Sample numbers and definitions of error bars are given in the figure legends.

## Supporting information

**S1 Fig. Loss of chitin does not alter the distribution of Zona Pellucida domain proteins.** (**A–C**) Horizontal cryosections of control adult retina (A) or retinas in which *kkv* RNAi (B) or *Chs2* RNAi is expressed with *IGMR-GAL4*, stained for ObstA (A′, B′, green in A, B), chitin (red in A, B, magenta in C) and Chp (blue in A, B, green in C). Obst-A expression is strongly reduced by *kkv* knockdown, and chitin levels and corneal lens shape appear wild-type when *Chs2* is knocked down. (**D, E**) 51 h APF retinas containing clones in which *kkv* RNAi is expressed with *IGMR-GAL4*, marked by GFP (green), stained for Dyl (D′, red in D) or Pio (E′, red in E), and Cad (blue). (**F–I**) Control retinas (F, H) and retinas in which *kkv* RNAi is expressed with *IGMR-GAL4* (**G, I**) at 51 h APF (F, G) or adult stage (H, I), showing Dpy-YFP expression

(**F″, G″,** green in **F–I**) and stained for chitin (red) and Cad (blue, **F, G**) or Chp (blue, **H, I**). Scale bars: 20 μm (**A–C, H, I**), 10 μm (**D–G**). *kkv* knockdown does not affect the level or distribution of Dpy-YFP on the apical side of the pupal retina or in the adult pseudocone.
(TIF)

**S2 Fig. Reb and Exp act redundantly on corneal lens morphology.** (**A–F**) Retinas stained for Cad (green) and Reb (**A′, B′, E′**, magenta in **A, B, E**) or Exp (**C′, D′, F′**, magenta in **C, D, F**). (**A, C, E, F**) control; (**B**) *IGMR>reb RNAi*; (**D**) *IGMR>expRNAi*. (**A–D**) 50 h APF; (**E, F**) 54 h APF. (**G–J**) *IGMR>rebRNAi* (**G, H**) or *IGMR>expRNAi* (**I, J**) 54 h APF retinas (**G, I**) or horizontal adult eye cryosections (**H, J**) stained for chitin (magenta) and Cad (green in **G, I**) or Chp (green in **H, J**). In the absence of Exp, chitin is lost at 54 h APF, except in the mechanosensory bristles, which do not express *IGMR-GAL4*. However, chitin expression recovers in the adult, indicating a contribution of Reb later in development. Scale bars: 10 μm (**A–F, G, I**), 20 μm (**H, J**). (**K, L**) Graphs showing the outer (**K**) and inner (**L**) angles between adjacent corneal lenses in adult eye horizontal cryosections for wild-type control (*n* = 80/6), *IGMR>reb RNAi* (*n* = 188/13), *IGMR>exp RNAi* (*n* = 134/18). The data underlying (**K, L**) can be found in S1 Data.
(TIF)

**S3 Fig. Loss of chitin from lattice cells has a minor effect on corneal lens architecture.** (**A, B**) 50 h APF retinas in which *kkv RNAi* is expressed in central cells with *spa-GAL4* in clones labeled with myrTomato (red, **A**) or in lattice cells with *54-GAL4* in clones labeled with anti-β-galactosidase (red, **B**), stained for chitin (**A′, B′**, green) and Cad (blue). Scale bars: 10 μm. (**C**) Graph showing corneal lens width in adult plastic sections or cryosections for wild-type control (*n* = 96/11), *IGMR>kkvRNAi* (*n* = 64/4), *IGMR>rebRNAi; expRNAi* (*n* = 35/3), *spa>kkv RNAi* (*n* = 71/4), *spa>rebRNAi; expRNAi* (*n* = 47/4) and *54>rebRNAi; exp RNAi* (*n* = 113/9). (**D**) Graph showing chitin fluorescence intensity in a 3 μm wide region at the adult corneal lens edge normalized to its center (see Fig 3I) in wild-type control (*n* = 79/11), *54>kkv RNAi* clones (*n* = 14/6) and *54>rebRNAi; expRNAi* (*n* = 113/9). **\*\****p* = 0.0048 (*54>kkv RNAi* v. control), **\*\****p* = 0.0052 (*54>reb, exp RNAi* v. control). Each point represents the mean value for one retina, and error bars show mean ± SEM. The data underlying (**C, D**) can be found in S1 Data.
(TIF)

**S4 Fig. Effect of excess chitin in the late pupal corneal lens.** (**A–F**) *w¹¹¹⁸* (**A, B**), *spa>UASkkvGFP* (**C, D**) or *54>UASkkvGFP* (**E, F**) 54 h APF retinas (**A, C, E**) or horizontal adult eye cryosections (**B, D, F**) stained for chitin (magenta) and Cad (green in **A, C, E**) or Chp (green in **B, D, F**). (**G, H**) Cryosections of 84 h APF retinas containing clones labeled with GFP (green, **G**) or myrTomato (red, **H**) in which *UAS-reb* is driven by *IGMR-GAL4* (**G**) or *spa-GAL4* (**H**), stained for chitin (red in **G**, green in **H**) and Chp (blue). (**G′, G″**) show enlargements of individual wild-type (**G′**) or *reb*-misexpressing (**G″**) corneal lenses stained for chitin with Calcofluor White. Scale bars, 10 μm (**A, C, E**), 20 μm (**B, D, F–H**), 2 μm (**G′, G″**). Blue arrows mark wild-type ommatidia and yellow arrows *reb*-expressing clones.
(TIF)

**S1 Data. Source data for all graphs.** The measurements for each individual corneal lens are given, grouped by retina. Means and standard deviations are shown at the bottom of each column, and the statistical tests and calculated *p* values are at the right of each sheet. Separate sheets show the data underlying Figs 2O, 2P, 3L, 4J, 4K, 4L, 4M, S2K, S2L, S3C, and S3D.
(XLSX)

## Acknowledgments

We thank Marta Llimargas, Anna Jaźwińska, Hélène Chanut, Matthias Behr, the Bloomington *Drosophila* stock center, the Vienna *Drosophila* resource center, the Kyoto stock center, and the Developmental Studies Hybridoma Bank for fly stocks and reagents. Information available on FlyBase was invaluable for this work. We thank NYULH DART

Microscopy Laboratory members Alice Liang, Jason Yin and Jason Liang for consultation and assistance with plastic sections, and this core is partially funded by NYU Cancer Center Support Grant NIH/NCI P30CA016087. The manuscript was improved by the critical comments of Gira Bhabha, Maria Bustillo, Holger Knaut, Sudershana Nair, and Pragati Sharma.

## Author contributions

**Conceptualization:** Neha Ghosh, Jessica E. Treisman.

**Data curation:** Neha Ghosh, Eva Rojo-Iost.

**Formal analysis:** Neha Ghosh, Jessica E. Treisman.

**Funding acquisition:** Neha Ghosh, Jessica E. Treisman.

**Investigation:** Neha Ghosh, Eva Rojo-Iost.

**Supervision:** Jessica E. Treisman.

**Writing – original draft:** Neha Ghosh.

**Writing – review & editing:** Eva Rojo-Iost, Jessica E. Treisman.

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
