## [Editor Report · Decision Letter 0]

13 Nov 2025

Dear Dr Treisman,

Thank you for submitting your manuscript entitled "Corneal lens curvature depends on localized chitin secretion" for consideration as a Short Report by PLOS Biology.

Your manuscript and the reports from Review Commons have now been evaluated by the PLOS Biology editorial staff, as well as by an academic editor with relevant expertise, and I am writing to let you know that we would like to proceed with considering your paper at PLOS Biology. We will send you a more detailed decision after you address the following requirements.

We first need you to complete your submission by providing the metadata that is required for full assessment. To this end, please login to Editorial Manager where you will find the paper in the 'Submissions Needing Revisions' folder on your homepage. Please click 'Revise Submission' from the Action Links and complete all additional questions in the submission questionnaire.

Once your full submission is complete, your paper will undergo a series of checks. After your manuscript has passed the checks we will send you further instructions. To provide the metadata for your submission, please Login to Editorial Manager (https://www.editorialmanager.com/pbiology) within two working days, i.e. by Nov 15 2025 11:59PM.

Kind regards,

Taylor

Taylor Hart, PhD,

Associate Editor

PLOS Biology

thart@plos.org

---

## [Editor Report · Decision Letter 1]

14 Nov 2025

Dear Dr Treisman,

Thank you for submitting your manuscript with reports from Review Commons to PLOS Biology. As mentioned previously, the PLOS Biology staff and the Academic Editor have evaluated your paper in light of the reports.

We see that the reviewers recruited through Review Commons expressed interest in your findings and wrote that the evidence presented was generally strong. As you noted, they asked for deeper explorations of some aspects of the study. After discussing these points and your revision plan with the Academic Editor, we would like to invite you to perform a Major Revision of your study. The Academic Editor also requested that you address some additional points as part of this revision (see below my signature).

**IMPORTANT - SUBMITTING YOUR REVISION**

1. A 'Response to Reviewers' file - this should detail your responses to the editorial requests, present a point-by-point response to all of the reviewers' comments, and indicate the changes made to the manuscript. Please include an updated R2R addressing the additional revisions after you complete them.

*Re-submission Checklist*

*Published Peer Review*

*PLOS Data Policy*

*Blot and Gel Data Policy*

Sincerely,

Taylor

Taylor Hart, PhD,

Associate Editor

PLOS Biology

thart@plos.org

Comments from the Academic Editor (lightly edited):

One point - not raised by the referees - concerns the Reb o/e expression experiments. The observation that increased Reb expression results in the formation of disorganized fibers raises two questions: what are the possible interpretations for the formation of these fibers? could it be that these fibers contribute to shape changes? Could it be that corneal lens curvature not only depends on localized chitin secretion (as stated in the title) but also on chitin (dis)organization?

Given the extent of revision needed, we cannot make a decision about publication until we have seen the revised manuscript and your response to the reviewers' comments. Your revised manuscript is likely to be sent for further evaluation by all or a subset of the reviewers.

---

## [Editor Report · Decision Letter 2]

6 Mar 2026

Dear Dr Treisman,

Thank you for your patience while we considered your revised manuscript "Corneal lens curvature depends on localized chitin secretion" for publication as a Short Report at PLOS Biology. This revised version of your manuscript has been evaluated by the PLOS Biology editors and the Academic Editor.

Based on our Academic Editor's assessment of your revision, we are likely to accept this manuscript for publication. Please also make sure to address the following data and other policy-related requests.

IMPORTANT: Please ensure that your next revision addresses all of the following editorial points:

**Title:

We think that it would be best to specify the study species in the title, as the findings are unlikely to generalize to vertebrate systems. Is the following alternative version acceptable to you?

"Corneal lens curvature depends on localized chitin secretion in Drosophila"

**Data:

You may be aware of the PLOS Data Policy, which requires that numerical data underlying the figures be provided. Please provide these data either within a new Supporting Information excel file S1 Data ("S1_Data.xlsx"), or as a permanent DOI’d deposition.

This applies to the following figure panels:

2OP

3L

4JKLM

S2KL

S3CD

Please then update the Data Location in the submission form, and also cite the location of the data clearly in all relevant main and supplementary Figure legends, e.g. “The data underlying this Figure can be found in S1 Data” or “The data underlying this Figure can be found in https://doi.org/10.5281/zenodo.XXXXX”

**Code availability:

Per journal policy, if you have generated any custom scripts or code during the course of this investigation, we require that you make it available without restrictions. Please ensure that the code is sufficiently well documented and reusable, and that your Data Statement in the Editorial Manager submission system accurately describes where your code can be found. Please also ensure that you choose a license for your code and include a Readme file.

We expect to receive your revised manuscript within two weeks.

*Published Peer Review History*

*Press*

Sincerely,

Taylor

Taylor Hart, PhD,

Associate Editor

thart@plos.org

PLOS Biology

---

## [Editor Report · Decision Letter 3]

11 Mar 2026

Dear Dr Treisman,

Thank you for the submission of your revised Short Report "Curvature of the Drosophila corneal lens depends on localized chitin secretion" for publication in PLOS Biology. On behalf of my colleagues and the Academic Editor, François Schweisguth, I am pleased to say that we can in principle accept your manuscript for publication, provided you address any remaining formatting and reporting issues. These will be detailed in an email you should receive within 2-3 business days from our colleagues in the journal operations team; no action is required from you until then. Please note that we will not be able to formally accept your manuscript and schedule it for publication until you have completed any requested changes.

As a minor note, we changed the filename of your source data file to "S1_Data.xlsx", in accordance with our required format.

PRESS

Sincerely,

Taylor

Taylor Hart, PhD,

Associate Editor

PLOS Biology

thart@plos.org